# The Relative Roles of Ambient Temperature and Mobility Patterns in Shaping the Transmission Heterogeneity of SARS-CoV-2 in Japan

**DOI:** 10.3390/v14102232

**Published:** 2022-10-11

**Authors:** Keita Wagatsuma, Iain S. Koolhof, Reiko Saito

**Affiliations:** 1Division of International Health (Public Health), Graduate School of Medical and Dental Sciences, Niigata University, Niigata 951-8510, Japan; 2Japan Society for the Promotion of Science, Tokyo 102-0083, Japan; 3College of Health and Medicine, School of Medicine, University of Tasmania, Hobart 7000, Australia

**Keywords:** SARS-CoV-2, transmissibility, ambient temperature, mobility patterns, epidemics

## Abstract

We assess the effects of ambient temperature and mobility patterns on the transmissibility of COVID-19 during the epidemiological years of the pandemic in Japan. The prefecture-specific daily time-series of confirmed coronavirus disease 2019 (COVID-19) cases, meteorological variables, levels of retail and recreation mobility (e.g., activities, going to restaurants, cafes, and shopping centers), and the number of vaccinations were collected for six prefectures in Japan from 1 May 2020 to 31 March 2022. We combined standard time-series generalized additive models (GAMs) with a distributed lag non-linear model (DLNM) to determine the exposure–lag–response association between the time-varying effective reproductive number (*R_t_*), ambient temperature, and retail and recreation mobility, while controlling for a wide range of potential confounders. Utilizing a statistical model, the first distribution of the mean ambient temperature (i.e., −4.9 °C) was associated with an 11.6% (95% confidence interval [CI]: 5.9–17.7%) increase in *R_t_* compared to the optimum ambient temperature (i.e., 18.5 °C). A retail and recreation mobility of 10.0% (99th percentile) was associated with a 19.6% (95% CI: 12.6–27.1%) increase in *R_t_* over the optimal level (i.e., −16.0%). Our findings provide a better understanding of how ambient temperature and mobility patterns shape severe acute respiratory syndrome coronavirus 2 (SARS-CoV-2) transmission. These findings provide valuable epidemiological insights for public health policies in controlling disease transmission.

## 1. Introduction

Since the first detection of the severe acute respiratory syndrome coronavirus 2 (SARS-CoV-2) in Wuhan, China at the end of 2019, the coronavirus disease 2019 (COVID-19) pandemic has been the defining global health crisis of our time and the greatest public health challenge, leading to political, social, and economic challenges and a devastating loss of lives [1,2,3]. According to data reported as of July 2022 by the World Health Organization (WHO), more than 550.0 million and 6.0 million people have been infected and died, respectively, worldwide [4]. In Japan, the first case of SARS-CoV-2 was confirmed on 15 January 2020, and since then, regional transmission has continued throughout the country. As the incidence began to increase rapidly, a state of emergency was declared by the government on April 16, 2020, requiring a voluntary reduction in physical contact, which was considered to have led to the control of the epidemic in 2020 [5,6,7]. In mid-March 2021, the number of patients increased again, and the fourth wave started with the Alpha (B.1.1.7) variant outbreak, which was eventually followed by an outbreak of the relatively high transmissibility Delta (B.1.617.2) variant, and then a sixth wave of widespread Omicron (B.1.1.529) variant transmission, which is still raging across Japan in 2022 [8,9,10].

The accumulated epidemiological evidence supports an association between SARS-CoV-2 transmission dynamics and meteorological drivers, which continues to gain attention. The mean ambient temperature has been identified as a consistent crucial driver; however, its relative importance is a matter of current debate, and other details of the association remain unexplored [11,12,13,14,15,16]. Indeed, a recent broader systematic review of 62 ecological studies reported a negative association between the ambient temperature and incidence in specific countries [17]. While multiple studies have investigated this association, some of the studies had methodological weaknesses and conflicting results [18,19]. Notably, the conflicting findings include positive [20,21], negative [22,23], nonlinear [24,25], and non-significant [26,27] associations. Such inconsistencies may be attributed to relatively small sample sizes and short observation periods, in addition to the fact that most previous epidemiological studies have considered a narrow range of only a few meteorological drivers. This limits their ability to detect associations between these meteorological drivers and the transmission dynamics of SARS-CoV-2 [13]. Indeed, most of these previous studies assumed a linear relationship between the ambient temperature and incidence, thus raising the question of whether extreme temperatures, such as cold and heat waves, also continuously increase the transmission risk.

In addition to crucial epidemiological evidence that the mean ambient temperature contributes to the transmission dynamics of SARS-CoV-2, digital proxies of human mobility patterns are likely to play a key role in directly inferring unobservable transmission patterns as a function of time [28,29,30,31,32,33,34,35,36,37,38,39,40]. A notable study by Kraemer et al. reported that mobility statistics (including notified COVID-19 data and real-time travel data) accurately recorded the SARS-CoV-2 transmission spread between Chinese cities. Notably, the findings revealed that the frequency of introduction from Wuhan predicted the size of the epidemic in other provinces [29]. Indeed, various mobility pattern datasets have become widely available during the ongoing COVID-19 pandemic and have been used to monitor time-dependent patterns of physical distancing [40,41,42]. It is likely that enhanced population mobility patterns play a relatively important role in modeling time-varying transmission dynamics, ranging from increased infections among humans to an increase in viral transmission at the local level (i.e., prefectures or cities).

Epidemiological studies of this topic face important challenges in modeling the potential environmental and behavioral drivers and transmission dynamics dependencies. According to previous systematic reviews on disease modeling, many of the methodological weaknesses could be attributed to inadequate study designs and statistical methods [43]. For instance, many studies have failed to consider the possibility of a non-linear relationship and temporally lagged effects of drivers and incidence. Notably, they did not account for time-varying nor location-specific confounders [13,44,45]. Time-series statistical regression modeling, which is often used to quantify short-term associations of environmental exposures with health outcomes, such as infectious diseases, would allow investigators to control for seasonality, long-term trends, other confounders, and autocorrelation, as well as explore the association of delayed exposure effects [46]. Despite the advantages of this method, its use has not been overly prevalent in research [13,44,45,47,48]. Additionally, many of these previous studies did not control for or only controlled for a few crucial potential confounders when describing the disease transmission dynamics, which include other behavioral and environmental drivers, the implementation of non-pharmaceutical interventions (NPIs), and temporal changes in population immunity.

Another crucial factor that may have affected the findings of previous studies is the choice of incidence (e.g., newly confirmed cases) as the main outcome of the models (i.e., dependent variable). Considering the relatively large number of undocumented infections and the non-uniform variations in temporal lagged structures between infection and symptom onset, and between testing and reporting, utilizing incidence as a dependent variable may not be an optimal disease modeling methodology for assessing the effects of behavioral and environmental drivers [13,44,45,49,50]. 

The transmission heterogeneity of infectious diseases should also be considered when formulating time-series statistical regression models. Generally, the observation of each case with a contagious disease is not independent, and this characteristic is referred to as a dependent happening [48]. In particular, the phenomenon of super-spreading SARS-CoV-2 events, owing to the heterogeneity (i.e., overdispersion) in the number of secondary transmissions, is a crucial epidemiological characteristic that leads to sustained transmission. A study in Hong Kong during the early stages of the pandemic found that only 19.0% of the infected people generated 80.0% of all transmissions [51]. In this context, methodological challenges for disease modeling include capturing and explaining the complex relationships between the potential driving factors that characterize time-varying transmission dynamics; this has also remained a crucial knowledge gap and is not based on a plausible empirical consensus.

In the present study, we analyzed the complex non-linear and temporally delayed associations between ambient temperature, mobility patterns, and time-dependent *R_t_* during approximately two SARS-CoV-2 epidemiologic years of the pandemic in Japan. Notably, we analyzed the readily accessible empirical data, adopted flexible time-series statistical modeling approaches, and adjusted for a wide range of potential confounders. We used time-varying *R_t_* to quantify the dynamics of transmission, while considering the impact of population immunity on disease transmission. Additionally, the changes in other meteorological drivers (i.e., relative humidity, precipitation, and wind speed), risk awareness through adherence to personal protective behaviors (e.g., wearing a mask or washing hands), implementation of specific NPIs (e.g., state of emergency declaration), emergence of new variants of concern, and ultimately vaccination campaigns were also incorporated into the model. Disentangling the roles of ambient temperature and community mobility patterns in the underlying mechanisms of transmission dynamics may provide an integrated characterization and understanding of the conclusions of previous studies.

## 2. Materials and Methods

### 2.1. Design Setting

This epidemiologic time-series statistical analysis assessed the nonlinear and delayed associations between the time-varying transmissibility of SARS-CoV-2 and behavioral and environmental drivers (i.e., mean ambient temperature and community mobility patterns) in Japan, in the context of an ecological study design. The study period ranged from 1 March 2020 to 31 March 2022 and roughly represented approximately two epidemiological years of the pandemic in Japan. Generally, Japan is located at latitudes of approximately 26° N to 43° N and longitudes of approximately 127° E to 141° E in the Western Pacific region and is constructed of a total of 47 Japanese prefectures (covering the whole country) from north to south (Appendix A).

### 2.2. Empirical Datasets

#### 2.2.1. Epidemiological Data

In the present study, we retrieved the prefecture-specific daily time-series of newly confirmed COVID-19 cases for six main Japanese prefectures (Hokkaido, Tokyo, Aichi, Osaka, Fukuoka, and Okinawa) over the study period (between 1 May 2020 and 31 March 2022) (Figure 1 and Appendix A). We selected these locations and periods based on geographical diversity and a relatively large number of cumulative confirmed, notified cases (cumulative confirmed, notified cases over 150) to reduce the effects of missing estimates and extreme stochastic variability of time-varying transmissibility over time. The dates of reports were systematically collected from the official COVID-19 case reports made available online by the Ministry of Health, Labor, and Welfare, Japan (MHLW) [52] and the open-source data provided by J.A.G. JAPAN Corporation (which summarize the press releases of confirmed COVID-19 cases published by local governments) [53]. The open-source data were supplemented and verified with information from the reporting agencies’ press releases. In the present study, we compiled confirmed, newly-notified COVID-19 cases published on these websites, identified the patients, and created an integrated dataset. In Japan, it is mandatory to report all laboratory-confirmed cases of COVID-19 with validated testing methods, including reverse transcriptase-polymerase chain reaction (RT-PCR) tests using respiratory samples or saliva, quantitative antigen tests using respiratory samples or saliva, and point-of-care antigen tests using respiratory samples. Public health centers are notified of newly detected COVID-19 cases, epidemiological and clinical information is retrieved from each recorded notification, and press release information is published on the website [54]. Note that all cases analyzed in this study were confirmed by RT-PCR.

#### 2.2.2. Meteorological Data

The Automated Meteorological Data Acquisition System (AMeDAS) developed by the Japan Meteorological Agency (JMA) is a high-resolution surface observation network for investigating meteorological conditions in Japan. In the present study, we retrieved prefecture-specific daily time-series of meteorological data for six selected Japanese prefectures collected by the AMeDAS over the study period [55]. Daily meteorological data, including mean ambient temperature (units: °C), relative humidity (units: %), precipitation (units: mm), and wind speed (units: m/s), published by the website were used as independent variables in the time-series statistical models presented here. Daily meteorological data collected from meteorological observatories (meteorological stations) situated in the prefectural capital city were utilized for each prefecture. Notably, these data are often used as the gold standard in the environmental epidemiology field [56].

#### 2.2.3. Mobility Data

To assess the dynamics of mobility patterns at a population level, in this present study, we retrieved prefecture-specific daily time-series from Google’s COVID-19 Community Mobility Reports’ data for selected six Japanese prefectures over the study period [57]. For Google’s mobility data, the measured mobility was reported for six data streams: grocery and pharmacy mobility (e.g., actively going to grocery markets, food warehouses, and drugstores), park mobility (e.g., actively going to national parks, public beaches, and marinas), residential mobility (e.g., actively going to the place of residence), retail and recreation mobility (e.g., actively going to restaurants, cafes, and shopping centers), transit station mobility (e.g., actively going to the metro and bus), and workplace mobility (e.g., actively going to the workplace and office). However, it is likely that these types of mobility data do not accurately describe associations with the transmission dynamics of SARS-CoV-2. Therefore, we utilized mobility patterns related to retail and recreation in our present analyses based on previous empirical domain knowledge of how this category likely represents the mobility in close-contact settings associated with a substantially increased transmissibility of SARS-CoV-2 [6,39,58]. Indeed, these mobility data can be utilized as an optimal proxy to reflect real-time trends in human movement patterns and human behavior. This measure quantifies the percentage deviation (units: %) from the median baseline for each day of the week for five weeks (between 3 January 2020 and 6 February 2020).

#### 2.2.4. Variants

The time of transmission of each SARS-CoV-2 variant was defined according to the literature [59]. Among the pandemic waves in Japan, the period from 16 January 2020 to the third wave involved the wild-type virus, the fourth wave was the Alpha variant (B.1.1.7), the fifth wave was the Delta variant (B.1.617.2), and the sixth wave was the Omicron variant (B.1.1.529). Although the duration of the wave and emergence of the first identified variant case differ depending on the prefecture, a uniform period has been adopted nationwide for each variant based on changes in the number of newly notified cases and the issuance period of the state of emergency declaration. Therefore, we defined the wild-type period as 16 January 2020 to the end of February 2021, the Alpha variant period from 1 March 2021 to the end of May 2021, the Delta variant period from 1 June 2021 to the end of November 2021, and the recent Omicron variant period from 1 December 2021 to the end of the study period.

#### 2.2.5. Vaccine Registry Data

To record dispatched and administered doses, the Japanese government has developed a vaccination record system (VRS). In the present study, we retrieved the prefecture-specific daily time-series of the administered dose data for six selected Japanese prefectures collected by the Digital Agency over the study period [60].

### 2.3. Statistical Analysis

#### 2.3.1. Descriptive Statistics

To clearly understand the temporal dynamics of the multitudes of epidemiological datasets in the present study, descriptive statistics were used. We visually assessed the trends of time-series variations in the number of new daily COVID-19 cases, meteorological variables (i.e., mean ambient temperature, relative humidity, precipitation, and wind speed), mobility patterns (i.e., retail and recreational mobility patterns), and the daily number of SARS-CoV-2 vaccinations during the study period (between 7 May 2020 and 7 June 2020) to assess the key characteristics of the dataset included in this study. We then described the probability distributions of the time-series data for the dependent and independent variables in the six selected Japanese prefectures using the following descriptive statistics: mean, SD, minimum (Min), 25th percentile (P_25_), 50th percentile (P_50_), 75th percentile (P_75_), and maximum (Max).

#### 2.3.2. Estimating Time-Dependent Transmissibility

To quantify the time-dependent transmissibility (transmission intensity) of SARS-CoV-2 throughout the epidemic, the effective reproductive number (*R_t_*) was estimated from the incidence time-series surveillance data by applying a simple branching process model in the context of Bayesian inference [61]. *R_t_* can be estimated by the ratio of the number of new infections generated at time step *t* (*I_t_*) to the level of infectiousness of the infected individuals at time *t*. This is calculated using the following equation: $\sum_{s=1}^{t}I_{t-a}w_{s}$, where the sum of infection incidence up to time step *t* − 1 is weighted by the infectivity function *w_s_*. *R_t_* is the average number of secondary cases that each infected individual would infect if the conditions remained as they were at time *t*; thus, *R_t_* is commonly used to characterize pathogen transmissibility during an epidemic. Generally, a *R_t_* value greater than one indicates that more cases are occurring and the infection is spreading, while a *R_t_* of less than one indicates that the spread of the infection is decreasing. Theoretically, we need information regarding the generation time, which is defined as the period between the infection of the index and the next case; however, this information is usually difficult to ascertain. Instead, the time-varying *R_t_* value can be adjusted to include the serial interval (SI), which is defined as the interval between the onset of the index and the next case, as an infectivity function, assuming a gamma distribution [61,62]. We utilized the SI (mean SI: 4.7 days, standard deviation [SD] SI: 2.9 days) described by Nishiura et al., and our time-varying estimates were made with a seven-day sliding window [63]. Statistical estimation was performed using R statistical programming software version 4.1.0 with the “EpiEstim” package. Detailed information on the mathematical framework for estimating the time-dependent transmissibility of SARS-CoV-2 and the selection of serial interval probability distribution and smoothing window are described in the Appendix A.

#### 2.3.3. Modeling Approaches

In the present study, we conducted a retrospective assessment of the time-dependent non-linear and delayed associations between daily *R_t_*, retail and recreation mobility patterns, and mean ambient temperature, utilizing a cutting-edge time-series regression analytic approach of statistical modeling. Specifically, we formulated a standard time-series generalized additive model (GAM) with a gamma probability distribution family and logarithmic-link probability function, allowing for overdispersion in the observational epidemiologic data, combined with distributed lag nonlinear models (DLNMs) [64]. Briefly, the general algebraic definition of time-series statistical models is formulated as follows, with further extensions attempted:$$P\left(Y_{t}\;|\;R_{t}\right)\;\sim \;Gamma\left[exp\left(\alpha +\sum_{\tau }f_{G}\left(x_{\tau ,t,i}\right)+\sum_{\mu }h_{G}\left(y_{\tau ,t,i}\right)+S_{t,i}+P_{i}+La_{i}+Lo_{i}+\sum_{i}f\left(z_{t}^{i};\theta \right)+D_{t}+H_{t}\;+V_{t}+\min\left(w_{limit,i},w_{t,i}\right)+O_{t}+ln\left(R_{t-1}\right)+\varepsilon_{t,i}\right)\right]$$
where *Y_t_* is the outcome time series; *R_t_* is the expected time series of the daily time-dependent effective reproductive number in prefecture *i* on day *t*; *α* corresponds to the overall intercept; and *f_G_*(x_τ,t,i_**) denotes the cross-basis function with exposure and lag effects modelled by a natural cubic spline function and a linear function of mean ambient temperature and retail and recreation mobility in prefecture *i* on day *t,* respectively. This function characterizes the complex relationship between transmissibility and these potential drivers by simultaneously accounting for the nonlinear and delayed dependencies typically found in this type of assessment. The notation *h_G_*(*y*_τ,t,i_**) represents a penalized smoothing spline function of relative humidity, precipitation, wind speed, and daily number of SARS-CoV-2 vaccinations, to control for nonlinear confounding effects (term of nonlinear). *S_t,i_* is a fixed effect indicator variable that takes the value of one during the state of emergency declaration period in prefecture *i* on day *t* and zero. Otherwise, *P_i_* denotes prefectural characteristics or regional variable indicators in prefecture *i*. *La_i_* and *Lo_i_* denote latitude and longitude variables in prefecture *i*. $\sum_{i}f\left(z_{t}^{i};\theta \right)$ denotes natural cubic splines of time (seven degrees of freedom per year). *D_t_* denotes the day of the week at *t*. *H_t_* denotes public holiday in day *t*, and *V_t_* is a fixed effect indicator variable of each variant on day *t*. In this statistical model, *w_t,i_* is the degree of risk awareness in prefecture *i* on day *t* [40]. This was graded by assuming that it was linearly associated with the smoothed number of newly reported cases, following a positive association between confirmed cases and risk perception found in a study from the United Kingdom (UK), using longitudinal data [65]. The effect of this variable *w_t,i_* was capped at a predefined upper limit (*w_limit,i_*) that corresponded to the government’s definition of the “highest alert level” incidence in Japan (i.e., 25 confirmed cases per 100,000 population in a week) [66]. According to this definition, the daily number of cases giving an upper limit for each prefecture *i* (*w_limit,i_*) was specified as 497 in Tokyo, 315 in Osaka, 270 in Aichi, 188 in Hokkaido, 182 in Fukuoka, and 52 in Okinawa. *R_t_*_−1_ denotes autoregressive terms of the first order, accounting for potential serial correlation. *O_t_* denotes the logarithm of the yearly population by prefecture as an offset term [67], and term *ε_t,i_* indicates the error term. To quantify the total contribution, independent effects, and relative importance of behavioral and environmental drivers, we included all variables in the same model. By including all variables of interest in the same regression equation, we strengthened the interpretation of the effects as independent and additive, based on accumulated empirical knowledge. A detailed description of the modeling process is provided in the Appendix A.

### 2.4. Ethical Considerations 

The present ecological modeling study in statistical inference analyzed publicly available data. As such, the epidemiological datasets utilized in this study were de-identified and fully anonymized in advance, and the analysis of publicly available data with no identifying information did not require ethical approval. This study was conducted in accordance with the Declaration of Helsinki (revised in 2013). 

## 3. Results

### 3.1. Descriptive Description 

In the present study, we included a total of 2,982,262 cases reported in six selected Japanese prefectures over the study period (Table 1). As of 31 March 2022, the mean number of daily newly confirmed cases per prefecture was 710 (range: 0–20,040). The mean daily *R_t_* averaged over the six selected prefectures and in the study period was 1.30 (range: 0.04−20.63). The overall range of the observed values varied between approximately −10.40 and 32.80 °C for the ambient temperature, 23.00 and 100.00% for the relative humidity, 0.00 and 231.50 for the precipitation, 1.52 and 17.90 m/s for the wind speed, −71.00 and 34.00% for the retail and recreation mobility, and 0 and 228,078 doses for the number of vaccinations. The geographical time-dependent transmissibility variation revealed that, in general, Okinawa had the highest *R_t_* averaged over the present study period and days, while Hokkaido had the lowest (Figure 2A–F, Appendix A). Southern prefectures were generally hotter than northern counties, and clear seasonal patterns were observed for the daily mean ambient temperature (Appendix A). However, the seasonal trends of the relative humidity, precipitation, wind speed, retail and recreation mobility, and daily number of vaccinations were less obvious (Appendix A).

### 3.2. Characterizing the Associations between SARS-CoV-2 Time-Dependent Transmissibility and Environmental and Behavioral Drivers

In the present study, we modelled the nonlinear and delayed associations of the mean ambient temperature and retail and recreational mobility with the daily SARS-CoV-2 *R_t_* utilizing standard time-series multivariable GAMs combined with DLNMs. We adjusted for spatiotemporal variations in time-dependent *R_t_* and potential measured confounders, as described in the Appendix A. Based on preliminary pooled estimates of the three-dimensional plots, we found evidence of nonlinear associations between time-dependent *R_t_* and ambient temperature and retail and recreation mobility over lags of 0–21 days (Figure 3A−C). For ambient temperatures in the range of approximately 10–25 °C, we found an approximately linear inverse ambient temperature–*R_t_* relationship, with lower ambient temperatures nonlinearly associated with increased time-dependent SARS-CoV-2 transmissibility (Figure 3B). The corresponding pooled cumulative relative risks (RRs) were 1.01 (95% confidence interval [CI]: 0.99–1.04) at 10.3 °C (25th percentile), and 0.97 (95% CI: 0.95–1.00) at 24.6 °C (75th percentile), respectively (Figure 3B, Table 2). Compared with the optimum ambient temperature (with reference to 18.5 °C), the first distribution of the mean ambient temperature (i.e., −4.9 °C) was associated with an 11.6% (95% CI: 5.9–17.7%) increase in *R_t_* (Figure 3B, Table 2). Regarding retail and recreational mobility, high levels were associated with an increased risk of transmissibility. More specifically, retail and recreation mobility were almost unrelated to transmissibility when the mobility was lower than approximately −50.0%; however, above this level, a nonlinear positive J-shaped association was observed between mobility and *R_t_* (Figure 3D). The pooled cumulative RRs were 0.96 (95% CI: 0.94–0.97) at a retail and recreation mobility level of −23.0%, and 1.05 (95% CI: 1.03–1.06) at −10.0%, with reference to −16.0% (Figure 3D, Table 2). More specifically, a retail and recreation mobility level of 10.0% (99th percentile) was associated with a 19.6% (95% CI: 12.6–27.1%) increase in *R_t_* over the optimal level (i.e., −16.0%) (Figure 3D, Table 2).

### 3.3. Further Investigations

The main findings described above were confirmed by repeating the series of sensitivity analyses utilizing alternative DLNM specifications and controlling for potential confounders to verify the robustness of our results. For the mean ambient temperature and retail and recreation mobility, changing the incorporation of the degree of freedom (d.f.) of the natural cubic spline of time from seven d.f. per year to three d.f. per year (Appendix A) or to 11 d.f. per year (Appendix A) revealed that the observed risk effect shape was substantially robust over the different parameterizations. However, we observed relatively lower risks when we incorporated 11 d.f. and higher risks when we incorporated three d.f. for the mean ambient temperature and retail and recreation mobility. The model uncertainties around the effect estimates slightly change the d.f. setting of the natural cubic spline of time. In addition, redefining the lag-response dimension using a natural cubic spline and three equally placed internal knots, we observed similar exposure–lag–response associations for the mean ambient temperature and retail and recreation mobility patterns (Appendix A). These series of sensitivity analyses confirm the robustness of the main analysis, and our main findings were largely insensitive. This reveals that the main time-series statistical model utilized in the present study adequately captured the nonlinear and delayed associations between time-dependent SARS-CoV-2 transmissibility, the mean ambient temperature, and retail and recreation mobility.

## 4. Discussion

Utilizing the estimated time-varying *R_t_* for six Japanese prefectures, we retrospectively assessed the nonlinear and delayed associations of mean ambient temperature and mobility patterns with the transmission dynamics of SARS-CoV-2, controlling for temporal and spatial trends and other potential confounders. Taken together, we found evidence of a modest non-monotonic association between the mean ambient temperature and retail and recreation mobility with *R_t_* over lags of 0–21 days. During the study period, we also found that a lower mean ambient temperature and higher retail and recreation mobility were significantly associated with an increased *R_t_*. More specifically, a mean ambient temperature that decreases from 18.5 °C (50th percentile) to −4.9 °C (1st percentile) was associated with an increase of 11.6% (95% CI: 5.9–17.7%) in *R_t_*. Meanwhile, an increase in retail and recreation mobility from —16.0% (50th percentile) to 10.0% (99th percentile) led to an increase of 19.6% (95% CI: 12.6–27.1%) in *R_t_*. Additionally, our present findings were robust and remained stable in a series of sensitivity analyses when the degrees of freedom of the natural cubic spline used in adjusting for time and the lag–response dimensions were changed. To the best of our knowledge, the present study is the first to objectively validate the potential mechanism underlying the associations among potential key drivers that influence SARS-CoV-2 transmission dynamics using an explicit time-series statistical modeling approach. This finding reveals that the *R_t_* temporal patterns could be largely explained by the ambient temperature and mobility patterns. In addition to highlighting the role played by behavioral and environmental drivers, our work suggests that there may be an additional mechanism involved in shaping time-dependent transmission heterogeneity that remains to be identified.

In line with our modeling study, it is noted that the present findings on the association between the mean ambient temperature and time-dependent SARS-CoV-2 transmissibility are largely consistent with those of previous studies; however, the potential mechanism underlying these associations remains unclear [11,12,13,14,15]. Indeed, a large time-series statistical modeling study that assessed 1,908,197 confirmed COVID-19 cases from 190 countries, including four major climate zones with a wide temperature range (from approximately −30.0 °C to 40.0 °C) as of 13 April 2020, also observed an overall inverse association between the mean ambient temperature and the incidence of SARS-CoV-2 infections, which supports the present findings [13]. In addition to these existing studies, a few precise epidemiological modeling studies have used time-dependent DLNMs to precisely assess the association between the time-dependent transmissibility of SARS-CoV-2 and environmental exposure, and consistent conclusions have been reached regarding the mean ambient temperature. One crucial study by Sera et al., which analyzed the transmission in 409 cities in 26 countries across the globe during the early stages of the pandemic, concluded that 2.4% of the *R_t_* variation was attributable to ambient temperature [45]. Another significant study in the USA investigated the attributable fractions of the mean ambient temperature at 3.7% [44]. Similarly, a study in Tokyo also reported a cumulative RR of 1.3 (95% CI: 1.1–1.7) at the first mean ambient temperature percentile (i.e., 3.3 °C), which provides empirical support for our findings [48]. Furthermore, the laboratory studies conducted by Chin et al. suggested that SARS-CoV-2 is stable at approximately 4.0 °C, but at approximately 70.0 °C (higher temperature), the virus inactivation time decreases to five minutes, supporting existing epidemiological findings [68]. Indeed, in addition to the host effect, the duration of the virus outside the body is also expected to be adversely affected by the inactivation of the virus, owing to the breakdown of the lipid layer at higher ambient temperatures [69,70]. It has also been speculated that the plausible reason for the strong association between increased ambient temperature and a higher risk of COVID-19 could be due to poor indoor ventilation in winter [13]. People generally stay indoors more during the winter, when doors and windows are closed to keep the indoor environment warm, resulting in poor ventilation. Studies have shown that aerosols from infected people can pose an inhalation threat, even at considerable distances, in enclosed spaces with poor ventilation [71,72]. In line with similar qualitative and quantitative evidence from existing epidemiological, ecological, and laboratory studies, our results suggest that during our study period in Japan, elevated mean ambient temperatures may have affected the attenuation of viral viability and partially suppressed SARS-CoV-2 transmission. However, the impacts of meteorological drivers (including the mean ambient temperature) on SARS-CoV-2 transmission dynamics might depend on climate zones and seasons. Thus, further investigations in different climate conditions are needed to understand the contributions of meteorological factors to varying phases of the pandemic, vaccinations, and variant distributions.

Importantly, it is worth noting that the association between mobility patterns and the time-varying transmissibility of SARS-CoV-2 was qualitatively and quantitatively consistent with the findings of previous studies and that some general rules of thumb could be deduced while clarifying the interpretation of these indicators. However, this is not the only plausible determinant. A study that investigated 52 countries (including Japan) by Nouvelles et al. found an association between transmission dynamics and human mobility, suggesting that in approximately 73.0% of the countries analyzed, the transmissibility of SARS-CoV-2, that is, the time-varying *R_t_,* typically decreased with initial mobility, indicating that mobility explained a significant proportion of the variation in transmission (median adjusted *R*^2^ 48.0%) [39]. Additionally, the authors concluded that in approximately 80.0% of countries, transmission and migration were decoupled after strict control measures were relaxed, suggesting that any change in these relationships would reduce the predictive capacity after relaxation (from a median adjusted *R*^2^ of 74.0% pre-relaxation to a median adjusted *R*^2^ of 30.0% post-relaxation). Another study by Li et al. linked data on six community mobility metrics from Google with data on *R_t_* from 330 local authorities in the United Kingdom (UK) between June 2020 and February 2021 to quantify the effects of changes in community mobility patterns [73]. Compared to other mobility metrics, increased visits to retail and recreational places were associated with a substantial increase in the weekly change in *R_t_* (1.05 [99.2% CI: 1.04–1.06] per 15.0% weekly increase compared with baseline visits). Similar findings were also reported by Deforche et al., who showed that decreased retail and recreational mobility patterns had the greatest contribution to reducing the severity of pandemics in 35 high-income countries [74]. Another study utilizing time-varying DLNMs, similar to those used here, also concluded that human behavior changes and government intervention played a far greater role in the spread of the virus than did meteorological conditions (i.e., while the *R*_t_ dropped by 0.08% for every 10.0 °C jump in the ambient temperature, early interventions were associated with a decrease of 0.28%) [45]. In Japan, a recent study noted that more than 75.0% of 11,342 respondents aged 20 to 64 years had practiced preventive measures (social distancing, handwashing, coughing etiquette, and immunity fortification) [75]. Although mobility restrictions were not that strict, the response of the population towards these governmental requests led to a significant reduction in the number of trips, with the number of inter-prefectural travel halved across the country compared to pre-pandemic conditions [76]. Changes in human behavior during the pandemic are a cognitive response to the immediate threat of COVID-19, which has an indirect effect on the reduction in disease transmission [77]. Another crucial contribution of the present study is the explicit modeling of the shape of exposure–lag–response associations between mobility patterns and time-varying transmissibility. While there are studies highlighting potential transmission due to mobility patterns, there are only a few in the literature that assess the intensity and thresholds of the effect. Interestingly, we found a non-linear positive J-shaped association between mobility and *R_t_*, partially describing the heterogeneity of the effect of transmission risk owing to human mobility change. These findings justify the need to focus on the intensity of community mobility patterns (especially non-essential retail and recreational activities) to reduce transmission. Indeed, one modeling study reported from Israel suggested that SARS-CoV-2 transmission does not spread evenly throughout the population, highlighting the effectiveness of local strategies to target individuals at local risk [78]. However, our findings do not focus on the size of the outbreak or the temporal extent of the dynamics, but rather on the identification of time-varying associations. Therefore, further studies that consider complex network structures (especially social contacts and mixing patterns) are needed [79].

Although the present study focused on the need to assess the potential effects of specific drivers on the transmission dynamics of SARS-CoV-2 in Japan, multiple caveats and certain assumptions must be made when interpreting the estimates and applying the present results to the assessments. It should be noted that the present study is a preliminary finding describing the nonlinear and delayed effects of location-specific ambient temperature and community mobility metrics on the time-varying *R_t_* within the context of the approximately two years’ worth of the pandemic in Japan based on the analysis of secondary data (i.e., observational data) and is regarded as a type of ecological study in statistical causal inference [80,81,82]. That is, our findings are likely vulnerable to confounding, and indeed, the biological mechanisms and natural histories behind both social/behavioral and biological/intrinsic factors (e.g., clinical course of patients in characterizing secondary transmission, increased viral replication, high frequency or dose of pathogen shedding, or some other unknown host–pathogen relationships) were not assessed. While these factors influence the accurate modeling of the true extent of disease infections and transmission within populations, using disease surveillance data allows for reliable temporal trends in disease dynamics to be modeled and used in public health decision making. In particular, the influence of individual exposure on infection risk cannot be inferred over a large geographical area of Japan, and does not lead to causality at the individual level. Studies published at the individual level can overcome this problem. Second, the present study utilized reporting dates instead of infection dates or symptom onset dates to estimate time-dependent *R_t_*, which may have introduced potential time-varying biases attributed to reporting delays. Therefore, our analyses are subject to uncertainty owing to variations in the models and a lack of precision in the estimated exposure–lag–response associations. However, our analysis considered the plausible delayed effect of cross-basis functions in order to address these potential biases. Third, the present study only included six Japanese prefectures in the context of a limited period, which may limit the generalizability of our findings, as they are not representative of results from different epidemic periods in different geographical regions. To further our study, there is a need to increase the number of geographical regions or the set of indicators, or provide additional data, such as individual-level data. Fourth, the present study utilized specific mobility data streams (i.e., retail and recreation mobility) as a surrogate measure of human mobility patterns using published data available in Google’s COVID-19 Community Mobility Reports. In Japan, it was reported that eating and drinking without mask wearing are associated with an increased risk of transmission, thus the government advised to refrain from eating in a group [83]. Retail and recreation are often associated with gatherings and eating, and thus these parameters are chosen to represent mobility data in this study. As a result, we did not assess other different metrics (e.g., workplace mobility) associated with transmission dynamics. Analyzing all mobility metrics could help to refine the components of NPIs by restricting certain activities that are shown to increase time-dependent *R_t_* substantially or relaxing activities that have a smaller effect on transmission [73]. Although it might well approximate the overall mobility level in each prefecture of Japan, this alternative measure did not reflect the actual effects of human mobility (i.e., not the actual number of movements of people in domestic regions of Japan) [57]. Indeed, these mobility metrics are based on Google service users who enabled their location history, so they do not represent the whole population, and the data are usually shared at the aggregate level and, therefore, are impossible to analyze by factors such as age and working status. Furthermore, we were limited by the mobility data (e.g., the absence of data that had origin–destination matrices) and were therefore unable to construct a mechanistic model that could help gain more insights into the interaction between mobility patterns and time-varying *R_t_*. Fifth, although the present study adjusted for corresponding epidemic periods of the SARS-CoV-2 variants and the daily number of administered vaccination doses at the population level as a fixed effect in the statistical model, endogeneity issues may have been caused by unmeasured potential confounders. Indeed, socioeconomic and demographic factors (e.g., age, sex, and social capital in terms of social epidemiology) may drive complex epidemic dynamics [84,85]. Additionally, other potential drivers, such as adaptive evolution among multiple variants of SARS-CoV-2, viral interference (i.e., one virus being prevented from multiplying by another), founder effects (i.e., variants introduced into populations with locally elevated transmission can increase in proportion at a national level without a bona fide transmission advantage), and immune escape were not explicitly considered as a context of the clinical/genomic datasets in the statistical model [86,87,88,89,90]. Considering these possible driving factors affecting the transmission dynamics of SARS-CoV-2, improving the stability of the modeling estimates is a crucial subject for future studies. Sixth, the present study utilized a fixed SI (i.e., mean SI: 4.7 days, SD SI: 2.9 days) that was estimated in the early stages of the pandemic to quantify the time-dependent *R_t_* of SARS-CoV-2 [63]. Of note, there is a little possibility that the use of a fixed SI affected the quantification of *R_t_*. According to the previous literature, one study described the mean SIs of 3.5 and 3.0 days for the Omicron variants, which is a little shorter than the tentative estimates reported previously and indicates that our estimates for *R_t_* may have a slight degree of uncertainty [91]. A mean SI of 2.2 days was also reported for an outbreak in South Korea [92]. Additionally, there are indications for a potentially different place of replication in the host and a different route of entry for the Omicron variant, which suggests a mechanism that accounts for a shorter serial interval and a shorter incubation period [93]. These short SIs make timely contact tracing more challenging, which has a negative impact on reducing onward transmission. Further data-driven modeling of the time-dependent transmission dynamics of SARS-CoV-2 will be the subject of our future studies, and we will extend the methods described above. Seventh, although meteorological drivers, including mean ambient temperature, relative humidity, precipitation, and wind speed, were utilized in the present analysis as explanatory variables in the time-series statistical models, other potential confounders (e.g., diurnal temperature ranges (DTR), sunshine hours, ultraviolet (UV) levels, and air pollutants (e.g., particulate matter 2.5 [PM_2·5_], nitrogen dioxide, ozone, and sulfur dioxide) may be associated with the spread of SARS-CoV-2 [14,27,94,95]. Particularly, absolute humidity (i.e., actual amount of water vapor in the air at a given ambient temperature) is widely used in epidemiological studies of respiratory infections, and further modeling studies are needed to quantify the relative contribution of this potential driver to the transmission dynamics of SARS-CoV-2 across Japan [96]. However, several previous studies have also suggested consistent associations between these meteorological variables (i.e., mean ambient temperature, relative humidity, precipitation, and wind speed), which suggests they may have a high explanatory power [11,12,13,14,15,16]. Eighth, because our present study focused on pooled associations between time-dependent *R_t_*, mean ambient temperature, and retail and recreation mobility, we did not fully characterize the association between the heterogeneous transmission of SARS-CoV-2 in specific prefectures, which is beyond the scope of this study. For instance, the increase in local COVID-19 clusters in remote prefectures of Japan should be considered, and the potential epidemiological impact on SARS-CoV-2 transmission dynamics may differ in certain regions [97]. To identify the actual causes of the geographical heterogeneity among prefecture outbreaks in Japan, we need to validate these methods using more detailed spatiotemporal modeling analyses at the prefecture level [98]. Our results reflect an overall trend that should be interpreted with caution. Ninth, our results do not consider temporal uncertainties other than variations in the optimal lag lengths (i.e., optimal delays in effect). Although concerns arise from plausible settings of the incubation period of SARS-CoV-2 prior to symptom onset in notified cases, substantial data describing biological processes, the natural history of the virus, and host reservoir population transmission dynamics have not yet been pursued in depth. Tenth, changes in the number of administered doses at the population level in Japan were considered as a fixed effect covariate in the construction of the statistical model, and the number of individual administered doses (e.g., first, second, or third doses [i.e., booster]) could not be considered. Importantly, one observational study from Israel reported that the rate of confirmed infections was lower in the booster group than in the non-booster group by a factor of 11.3% (95% CI: 10.4–12.3) [99]. Therefore, it should be noted that this evidence could have been a source of bias in the estimated transmission patterns. Indeed, we did not model the waning of vaccine effectiveness throughout the epidemic wave, but tested lower effectiveness values. Eleventh, the present study attempted to focus on a time-series analysis of the entire approximately two years of the pandemic across Japan. Generally, the genetic variability of SARS-CoV-2 is large, and it is possible to meticulously identify the potential drivers affecting the mode of transmission by repeating the analysis of the effects of different regions on each outbreak (e.g., *R_t_* > 1.00) [100]. An analysis focused on such knowledge gaps will be addressed in our future modeling studies. Twelfth, reporting bias is always a problem when estimating the *R_t_* but may not have been fully eliminated in this analysis. The testing policy for RT-PCR testing did not change significantly over the study period, but the frequency and scale of screening tests may have been influenced by the scale of the epidemic [101]. However, we attempted to address this bias as far as possible through a repeated sensitivity analysis of the model. Finally, the underlying mechanisms that characterize the associations between transmission dynamics, meteorological drivers, community mobility patterns, and their epidemiological consequences in the context of the multifactorial nature of the epidemic process have not yet been fully quantified, which poses methodological challenges and data limitations for modeling [45,102,103,104]. Indeed, our findings need to be put into the context of complex uncertainties surrounding the characteristics of the novel virus, such as the incomplete knowledge of possible underlying mechanisms between weather conditions and the virus itself, the role of host immunity, and the potential influence of weather-sensitive human behaviors, such as indoor gatherings. Additionally, owing to the novelty of the virus, with less than approximately three epidemiologic year cycles of data available for most places, it is relatively difficult to fully disentangle seasonal signals or inter-annual trends from meteorological drivers and mobility patterns using modeling. These crucial limitations of an ecological study represent difficulties in the post hoc causal analysis of mass interventions that were implemented without a built-in evaluation design, such as randomization. Therefore, more broadly, the formulation of our time-series statistical model should be considered in light of the considerable uncertainty inherent in estimating the driver effects.

Finally, further prospective design analyses that describe the time-dependent transmissivity of infectious diseases attributable to changes in mobility patterns and meteorological variables are critical to confirm our main findings and establish definitive causality. As the pandemic continues to evolve, it will be necessary to repeat our modeling analysis by including data from genomic surveillance, seroprevalence surveillance, and broader respiratory viral surveillance to understand the effect of drivers on virus transmission after vaccination. Nevertheless, by utilizing only the available prefectural-level observational epidemiological data in Japan, we found that an ecological time-series modeling design could be exploited to examine the collective (i.e., population) impact of changes in behavioral and environmental drivers. Therefore, we believe that our approach will shed light on the assessment of the combined effects of drivers in Japan. 

## 5. Conclusions

While more meticulous follow-up studies that supplement our findings are needed, we believe that the present study disentangles the empirical evidence of a nonlinear and delayed temporal association between time-dependent SARS-CoV-2 transmissibility and the mean ambient temperature and mobility patterns during approximately two years of the pandemic across Japan. A more detailed picture of the complex associations between environmental and behavioral drivers and disease dynamics will contribute to an accurate evaluation of the consequences of heterogeneous transmission. This should also help us better understand the intensity and thresholds of indicators used to design appropriate public health strategies implemented at the individual or population level. Interestingly, most of the associations that we found were consistent with the accumulated theoretical mechanisms that link behavioral and environmental driving factors to transmission. In practice, this study contributes to the understanding that the effects of ambient temperature and human mobility act as substantial factors that govern (at least partially) the observed epidemic history of the pandemic in Japan. Furthermore, these findings will have implications for subsequent policies or warning strategies in light of ongoing or future pandemics. Nevertheless, there remains a lack of convincing quantitative evidence to support the complex association between the key driving factors that underlie the transmission dynamics of SARS-CoV-2. We strongly argue that the concerted efforts of comprehensive multi-city, multi-county collaborative studies are needed to guide the development of effective public health interventions. Moreover, these efforts should be expanded to the design of future widespread emerging and re-emerging infectious disease control measures and to the global assessment of the pandemic’s disease burden, in light of the progression of the more complicated transmission landscape. Explicit inferential modeling (including time-series statistical and mathematical models) that can realistically describe the behavior of infectious diseases based on intrinsic and extrinsic assumptions to describe the dynamics of pathogen transmission in populations during and after the pandemic critically needs to be considered in detail in future modeling studies. Altogether, these associations must be elucidated using additional concerted studies with different epidemiological designs that aim at assessing the Hills criteria of causation [105,106] so that any possible countermeasures, including alternative prescriptions, can be considered in the future. Finally, it is crucial to accurately study statistical causal inference in the context of infectious disease epidemiology in Japan, and our ultimate goal is to enhance future disease surveillance warnings with quantitative probabilistic inferences of disease risk rather than through risk assessments based on expert opinions of probable health outcomes. 

## Figures and Tables

**Figure 1 viruses-14-02232-f001:**
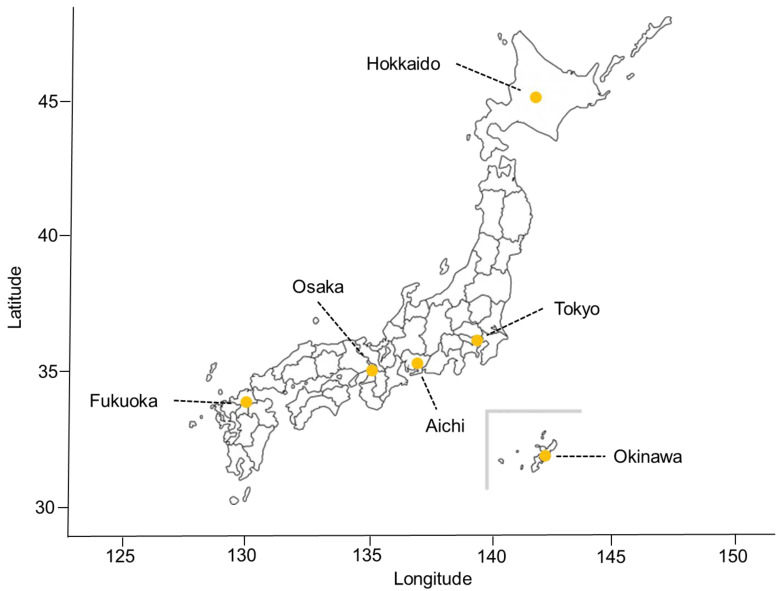
The geographic locations of the six selected Japanese prefectures. Japan is located at latitudes from approximately 26° N to 43° N and longitudes from approximately 127° E to 141° E in the Western Pacific region and constructed of a total of 47 Japanese prefectures (covering the whole country) from north to south: Hokkaido, Aomori, Iwate, Miyagi, Akita, Yamagata, Fukushima, Ibaraki, Tochigi, Gunma, Saitama, Chiba, Tokyo, Kanagawa, Niigata, Toyama, Ishikawa, Fukui, Yamanashi, Nagano, Gifu, Shizuoka, Aichi, Mie, Shiga, Kyoto, Osaka, Hyogo, Nara, Wakayama, Tottori, Shimane, Okayama, Hiroshima, Yamaguchi, Tokushima, Kagawa, Yamanashi, Nagano, Gifu, Shizuoka, Aichi, Mie, Shiga, Kyoto, Osaka, Hyogo, Nara, Wakayama, Tottori, Shimane, Okayama, Hiroshima, Yamaguchi, Tokushima, Kagawa, Ehime, Kochi, Fukuoka, Saga, Nagasaki, Kumamoto, Oita, Miyazaki, Kagoshima, and Okinawa (Appendix A). The orange dots indicate the geographical locations studied in selected six Japanese prefectures (i.e., Tokyo, Osaka, Aichi, Hokkaido, Fukuoka, and Okinawa).

**Figure 2 viruses-14-02232-f002:**
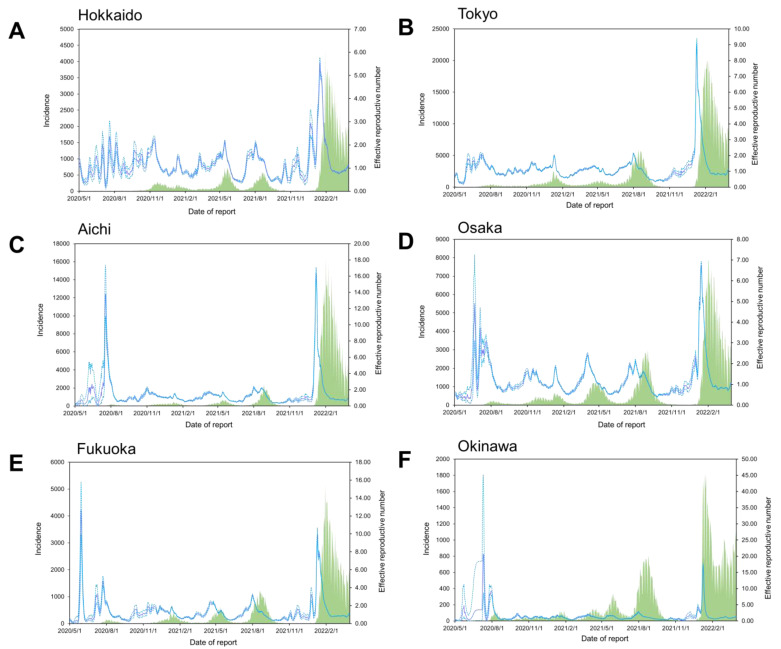
Epidemiologic curves for daily confirmed new COVID-19 cases, along with time-dependent effective reproductive numbers in Hokkaido, Tokyo, Aichi, Osaka, Fukuoka, and Okinawa, Japan. The number of coronavirus disease 2019 (COVID-19) cases by date of report (in green bars) from 1 May 2020 to 31 March 2022 in the six selected Japanese prefectures: (**A**) Hokkaido, (**B**) Tokyo, (**C**) Aichi, (**D**) Osaka, (**E**) Fukuoka, and (**F**) Okinawa. Purple lines and Blue dotted lines indicate the estimated time-varying effective reproductive numbers (*R_t_*) of severe acute respiratory syndrome coronavirus 2 (SARS-CoV-2) and the 95% confidence intervals (CIs), and the green bars represent daily incidence.

**Figure 3 viruses-14-02232-f003:**
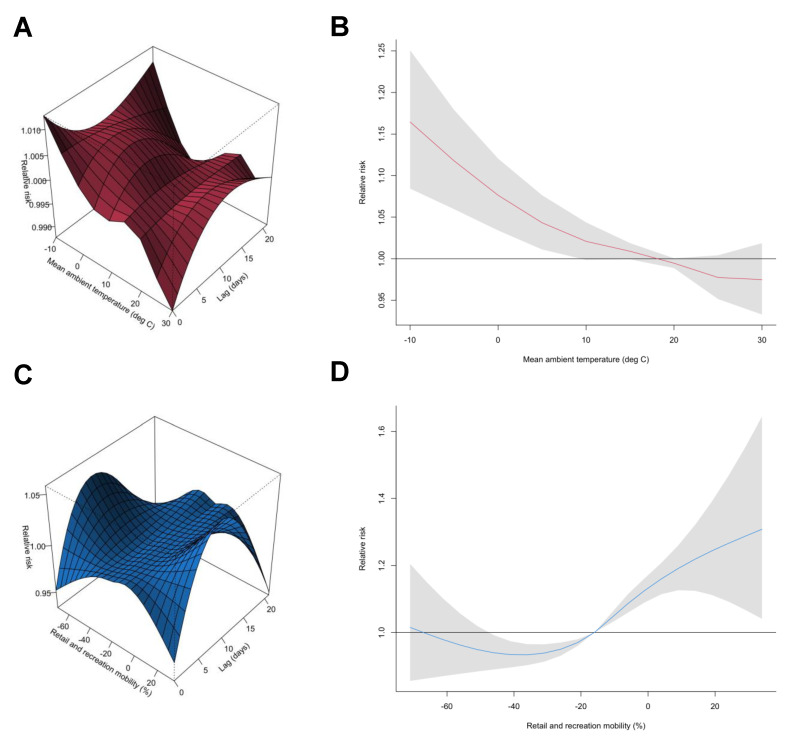
Visualizing pooled non-linear and delayed associations of the relative risks of time-dependent SARS-CoV-2 transmissibility with mean ambient temperature and retail and recreation mobility. (**A**) Three-dimensional plot of the association between daily mean ambient temperature (units: °C) and the percent change in estimated time-dependent effective reproductive number (*R_t_*) over a lag of 21 days. (**B**) Overall associations of the 21-day cumulative risk of the percent change in the estimated time-dependent *R_t_* of SARS-CoV-2 with daily mean ambient temperature (units: °C). (**C**) Three-dimensional plot of the association between daily retail and recreation mobility (units: %) and the percent change in estimated time-dependent *R_t_* over a lag of 21 days. (**D**) Overall associations of the 21-day cumulative risk of the percent change in estimated time-dependent *R_t_* of SARS-CoV-2 with daily retail and recreation mobility (units: %). Red and blue lines represent the estimated cumulative relative risks of time-dependent SARS-CoV-2 transmissibility, with shaded bands as 95% CIs. The corresponding reference levels were 18.5 °C and −16.0%, respectively. The present study covers the period between 1 May 2020 and 31 March 2022 in six selected prefectures (i.e., Hokkaido, Tokyo, Aichi, Osaka, Fukuoka, and Okinawa) in Japan. The relevant selected estimates (i.e., relative risks and 95% confidence intervals) for this figure are provided in Table 2.

**Table 1 viruses-14-02232-t001:** Descriptive statistics for the number of daily new confirmed COVID-19 cases, effective reproductive numbers, meteorological variables, and mobility patterns across all selected six prefectures over the study period.

Potential Drivers	Mean	SD	Min	P_25_	P_50_	P_75_	Max
Daily new confirmed cases	710	2038	0	21	92	393	20,040
Effective reproductive number	1.30	1.40	0.04	0.72	0.99	1.42	20.63
Mean ambient temperature (°C)	17.18	8.95	−10.40	10.30	18.55	24.60	32.80
Relative humidity (%)	70.27	12.95	23.00	61.00	70.00	79.00	100.00
Precipitation (mm)	5.02	14.97	0.00	0.00	0.00	2.00	231.50
Wind speed (m/s)	3.23	1.52	1.00	2.20	2.80	3.90	17.90
Retail and recreation mobility (%)	−16.66	10.36	−71.00	−23.00	−16.00	−10.00	34.00
Daily number of vaccinations (doses)	19,044	34,155	0	0	41	25,336	228,078

Abbreviations: SD, standard deviation; Min, minimum; P_25_, 25th percentile; P_25_, 50th percentile; P_75_, 75th percentile; Max, maximum. Notes: The present study covers the period between 1 May 2020 and 31 March 2022 in six selected prefectures (i.e., Hokkaido, Tokyo, Aichi, Osaka, Fukuoka, and Okinawa) in Japan.

**Table 2 viruses-14-02232-t002:** Assessing the specific relative risks of non-linear and delayed associations between the time-dependent transmissibility of SARS-CoV-2 and mean ambient temperature and retail and recreation mobility.

Potential Drivers	Lag (Days)				
	0	7	14	21	0−21
	RR (95% CI)	RR (95% CI)	RR (95% CI)	RR (95% CI)	RR (95% CI)
Mean ambient temperature (°C)					
−4.9 °C	1.00 (0.98, 1.02)	1.00 (0.99, 1.01)	1.00 (0.99, 1.01)	1.00 (0.98, 1.01)	1.11 (1.05, 1.17)
10.3 °C	0.99 (0.99, 1.00)	1.00 (1.00, 1.00)	1.00 (0.99, 1.00)	0.99 (0.98, 0.99)	1.01 (0.99, 1.04)
24.6 °C	0.99 (0.98, 1.00)	1.00 (0.99, 1.00)	0.99 (0.98, 1.00)	0.99 (0.99, 1.00)	0.97 (0.95, 1.00)
30.9 °C	0.98 (0.97, 0.99)	1.00 (0.99, 1.00)	1.00 (0.99, 1.00)	0.99 (0.98, 1.00)	0.97 (0.92, 1.02)
Retail and recreation mobility (%)					
−46.0%	0.98 (0.96, 0.99)	1.00 (0.99, 1.00)	1.00 (0.99, 1.00)	0.98 (0.97, 1.00)	0.93 (0.88, 0.99)
−23.0%	0.99 (0.99, 1.00)	0.99 (0.99, 0.99)	0.99 (0.99, 0.99)	1.00 (1.00, 1.00)	0.96 (0.94, 0.97)
−10.0%	1.00 (0.99, 1.00)	1.00 (0.99, 1.00)	1.00 (1.00, 1.00)	1.00 (1.00, 1.00)	1.05 (1.03, 1.06)
10.0%	0.99 (0.98, 1.00)	1.01 (1.01, 1.02)	1.01 (1.00, 1.01)	0.99 (0.98, 0.99)	1.19 (1.12, 1.27)

Abbreviations: RR, relative risk; CI, confidence interval. Notes: The present study covers the period between 1 May 2020 and 31 March 2022 in six selected prefectures (i.e., Hokkaido, Tokyo, Aichi, Osaka, Fukuoka, and Okinawa) in Japan. The temperatures −4.9 °C, 10.3 °C, 18.5 °C, 24.6 °C, and 30.9 °C correspond to the 1st, 25th, 50th, 75th, and 99th percentiles of mean ambient temperature, respectively; moreover, −46.0%, −23.0%, −16.0% −10.0%, and 10.0% correspond to the 1st, 25th, 50th, 75th, and 99th percentiles of retail and recreation mobility, respectively. Associations between the estimated time-varying effective reproductive number (*R_t_*) and mean ambient temperature (units: °C) and retail and recreation mobility (units: %) were described as relative risk (RR) with its 95% confidence intervals (CIs), with reference to 18.5 °C and −16.0%, respectively.

## Data Availability

An anonymized dataset that enables replication of the analysis is publicly available and is available from the corresponding author upon request.

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
