# Peer review of "The Relative Roles of Ambient Temperature and Mobility Patterns in Shaping the Transmission Heterogeneity of SARS-CoV-2 in Japan"

_viruses, 2022, doi:10.3390/v14102232_

Round 1

Reviewer 1 Report

The study examined the association of ambient temperature and mobility patterns on COVID-19 transmissibility in Japan. This is one of the very few studies in looking at both significant determinants on disease spread. Becasue of it, the article should be of interest to various readers in studying infectious disease dynamics and environment epidemiology. Some comments:

1. In abstract, what are the mobility proxies should be described.

2. Introduction is too long, suggest to be shortened.

3. Section 2.2.1: Any cases were confirmed by RAT, instead of PCR? Any possibility of underreporting?

4. Section 2.2.2: Absoulte humidity has been used widely in relating to respiratory infections (e.g. 10.1016/j.scitotenv.2019.134727). Why it was not been studied? 

5. Why retail and recreation was used, instead of the others such as workplace mobility? Tranmissions are more likely to occur in settings like workplace.

6. Different variants would have different SI. Has this been accounted?

7. Model equation is suggested to be showed in main text.

8. Section 3.1 is too long.

Reviewer 2 Report

The manuscript by “Wagatsuma et al.” reports the assess the effects of ambient temperature and mobility patterns on transmissibility during the epidemiological years of the pandemic in Japan. They found that distribution of the mean ambient temperature and retail and recreation mobility were significantly associated with an increase in Rt. These findings not only provide a better understanding of how ambient temperature and mobility patterns shape SARS-CoV-2 transmission, but also provide valuable epidemiological insights for public health policies in controlling disease transmission.  The study is innovative and the manuscript is well-written.

Minor comments:

Pathogen is one of the most important factors affecting the spread of epidemic diseases. As coronavirus, SARS CoV-2 has the characteristics of high variation. Therefore, insteading of analyzing the data of almost two years totally, analyzing the impact of different regions and various factors on the transmission of COVID-19 of each outbreak (Rt>1) can more accurately identify the factors that significantly affect the transmission mode of COVID-19.

Round 2

Reviewer 1 Report

The authors have addressed well to the comments.

Reviewer 2 Report

The authors have addressed well to the comments.